# The performance of restricted AIC for irregular histogram models

Sahika Gokmen [1,2] *, Johan Lyhagen [2]

**1** Department of Econometrics, Ankara Haci Bayram Veli University, Ankara, Turkey, **2** Department of Statistics, Uppsala University, Uppsala, Sweden

☯ These authors contributed equally to this work.
* sahika.gokmen@statistics.uu.se

**Data Availability Statement:** Tha analysis mainly include Monte Carlo study. The real data example underlying the results presented in the study are refereed in the text. Also, all the datasets and codes used in the study are available in the following link with DOI:10.5281/zenodo.8059567: https://zenodo.org/record/8059567.

## Abstract

Histograms are frequently used to perform a preliminary study of data, such as finding outliers and determining the distribution's shape. It is common knowledge that choosing an appropriate number of bins is crucial to revealing the right information. It's also well known that using bins of different widths, which called unequal bin width, is preferable to using bins of equal width if the bin width is selected carefully. However this is a much difficult issue. In this research, a novel approach to AIC for histograms with unequal bin widths was proposed. We demonstrate the advantage of the suggested approach in comparison to others using both extensive Monte Carlo simulations and empirical examples.

## Introduction

Histograms, introduced by Pearson (1985), are widely used for both density estimator and data presentation. This kind of a graphical representation of data can help to reveal information which cannot be discovered from standard summary statistics. Histograms are a helpful tool that is typically computationally straightforward and widely accessible, for instance in the majority of statistical computer applications.

In essence, a histogram's columns, also known as "bins", show the frequencies of the range of their base [1]. Based on this, an equal bin width histogram has two parameters: the bin width $h$ and the bin origin $x_0$. Accordingly, the intervals of bins are defined as $[x_0 + mh, x_0 + (m + 1)h]$ for integers $m$ [2]. The information of data stored in the histogram depends on the bin width and can be either too much or too little. For instance, larger $h$ leads to hide valuable information while smaller $h$ shows the data too random. These issues, which are referred to as undersmoothing and oversmoothing, respectively, prevent the histogram from revealing the pertinent information that it is capable of doing [2, 3].

The oversmoothing and undersmoothing problems make it more important to determine the number of bins (or, more precisely, the ideal bin width). Data driven methods for determining the number of equal width bins includes e.g. [3–10]. More advanced methods have been proposed which depends on the underlying distribution [11–14], cross validation [15, 16] or Bayesian theory [17, 18]. In addition to these techniques, Taylor (1987) made a significant advance to histogram analysis by determining the ideal bin width using the Akaike

**Funding:** Sahika Gokmen is supported by the International Postdoctoral Research Scholarship Program of The Scientific and Technological Research Council of Turkey (TUBITAK BIDEB 2219). The funder had no role in study design, data collection and analysis, decision to publish, or preparation of the manuscript.

**Competing interests:** The authors have declared that no competing interests exist.

Information Criterion (AIC). The purpose of the method is to estimate the best bin width through the maximization of likelihood function [19]. Taylor (1987) demonstrated that the performance of AIC is better than other data-based methods under the equal bin width condition. This originated AIC to be used in relation to histograms in different contexts [9, 10, 20–27].

All of the aforementioned strategies take only equal bin width into account. It can be far more useful in some circumstances to have various bin widths, though. For instance, Scott and Scott (2008) claimed that creating equal bin width histograms can be misleading when attempting to comprehend the underlying density information because the majority of frequency tables and the histograms that correspond to them are constructed using unequal widths [28]. Similar to this, Scott (2010) reported that using equal bin widths can result in bins that are too close to the modes [29]. However, only a small amount of study has examined at unequal bin width for histograms because it is more challenging to determine an unequal bin width than an equal bin width in practice. Amongst unequal bin width studies, Davies et al. (2009) argued that the reason for the rare unequal bin width preference is due to the decision theory framework. Accordingly, in this setting, there are many regular histogram procedures and these procedures are more difficult to adjust to unequal bin width in histograms (in other words irregular histograms) [10]. Having automatic procedures, computationally faster methods and having a variety of methodologies turns to an advantage to equal bin width histograms against the cost of getting the true distribution wrong. Some research which takes this issue into account is highly technical, like Engel (1997). In his paper, an adaptive version of multiresolution analysis for histogram (referred as multiresolution histogram) is proposed and the benefits of choosing unequal bin width are stressed [30]. Moreover, Barron et al. (1999) developed a methodology based on risk bounds which is used for model selection via penalization. This is applied on unequal bin width histograms but it is a purely theoretical exercise [31]. In a sequence of papers, Kanazawa has made important contributions. He derived optimality in the construction of histograms for both equal and unequal bin sizes [32, 33]. Furthermore, AIC is the focus in one of his papers [21] but not in the context of unequal bin widths. Rissanen and Speed (1992) extend the minimum description length principle to an unequal bin width histogram for density estimation by minimizing the largest absolute deviation between the histogram and its density [34]. Here it should be noted that the estimation of the density function is given a lot of attention in the majority of publications on unequal bin width histograms.

Besides the all literature, studies on penalized methods that account for unequal bins, such as AIC, are scarce [26, 27, 35]. Only Rozenholc et al. (2010) are regarded as AIC in this context out of the novel and aforementioned studies [26]. Their study aimed to combine equal and unequal histograms and may be considered as a comprehensive study as they examined the performances of 15 methods. The methods used are included in R-package *histogram* [36]. When the literature for histograms with different bin widths is examined, it has been realized that a technical improvement is restricted about determining the bin width. Therefore, the primary motivation of this research is to introduce the restricted AIC method which is a modified approach by using AIC for histograms with different bin widths. It has been thought that improving the AIC in this field, may increase its usage area.

Based on the aim of this study, the paper is structured as follows: in Section 2, the binomial model and the corresponding AIC is derived for unequal bins. Then, in Section 3, Monte Carlo experiments compare the proposed AIC method with other unequal bin width approaches. In Section 4, empirical examples are examined through the proposed AIC method and the other methods used in Section 3. Finally, Section 5 contains the conclusions.

## Determining the bin size of a histogram with unequal bin sizes via restricted AIC

Konish and Kitagawa (2008, p. 78) note that a histogram with $k$ bins can be seen as multinomial distribution. Let $n_j$ denote the number of observations in bin $j$ and $n = \sum_{j=1}^{k} n_j$ the total number of observations then the likelihood is

$$l(p_1, \ldots, p_k) \quad = \quad C + \sum_{j=1}^{k} n_j \ln (p_j)$$

where $C = \ln(n!) - \sum_{j=1}^{k} \ln(n_j!)$. The maximum likelihood estimates are $\hat{p}_j = n_j/n$. Then it is easily seen that

$$\text{AIC} \quad = \quad -2\left( C + \sum_{j=1}^{k} n_j \ln(\hat{p}_j) \right) + 2(k - 1).$$

Konish and Kitagawa (2008, p. 78) also noted that if two bins were merged, say bin 1 and 2, then the maximum likelihood estimator is

$$\hat{p}_1 \quad = \quad \hat{p}_2 = \left(\frac{n_1 + n_2}{2n}\right).$$

First we need to decide upon the maximum number of bins. We set it to $k = \lfloor n/\log(n) \rfloor$ (where $\lfloor x \rfloor$ denotes the largest integer not larger than $x$) as for example in [26]. The thresholds for the bins are created such that the bins has equal widths. In the first step, we compare all possible merges of bins that will end up in two bins. The best one is the one that minimizes the AIC and we select that one as our model if there were only two bins. There are $k - 1$ of possible ways to merge them. If we would like to merge into three bins then there would be $(k - 1)(k - 2)/2$ possible ways to merge. In general there are $\Pi_{i=1}^{K} (k - i)/K!$ ways to divide $k$ bins into $K$ groups. This becomes computationally very expensive. Our proposal is to fix the threshold given when considering two bins for the case of three bins. Then the best three bins model is compared to the best two bins model. If the best three bins model has a better AIC then continue by fixing also that threshold when considering four bins. Continue until the next number of bins does not improve AIC. This procedure is simple, computationally feasible and relies on sound statistical theory.

## A Monte Carlo experiment

The Monte Carlo simulation is made for analyzing the properties of proposed methods under a known setting. The setup is based on the previous studies, e.g. [10, 26]. The distributions included are: N(0,1), exp(1), U(0,1), a distribution where half of the observations are from N(0,1) and half from N(4,1), which we denote 0.5N(0,1)+0.5N(4,1), a $t$-distribution with 3 degrees of freedom and a Beta(1.5,1.5). The sample sizes are $n = (50, 100, 200, 400, 800)$ and the number of replicates is 5000 for each scenario. To discuss the performances of methods, loss functions are needed. We use the Hellinger distance, the $L_1$ and $L_2$ norms, between the true density and the estimated density as the results from previous studies show that the choice of the loss function is important for the results. The simulation is carried out in the R program [37] and all methods we compare with are implemented in the *histogram* package [36], except for TS where the *ftnonpar* package was used [38].

We compare our method with the 9 methods used for determining bin width for the unequal histogram in the [26] study. Further information on the methods compared and their

abbreviations are given in the Supporting information. The simulation results displayed in Tables 1–3. Table 1 demonstrates the results based on the Hellinger distance while Tables 2 and 3 demonstrate the results based on $L_1$ and $L_2$. The values in the tables show the relative distances compared to our AIC, which is abbreviated as AICr. This implies that a value above one indicates that the distance of the method was larger than AICr.

Simulation results for common distributions are consistent with Davies et al. (2009) and Rozenholc et al. (2010). When the comparison approaches are reviewed first, Hellinger distance is defined as a pessimist while $L_1$ and $L_2$ are defined as optimists for AICr. Also, there is an almost full agreement when we compare the number of cases $L_1$ and $L_2$ agreed in being larger or smaller than 1. Hence, the qualitative results are basically the same and we conclude that it doesn't matter if $L_1$ or $L_2$ is used. However, there are some notable differences between the Hellinger distance and them. AICr is the best method in 60.4% according to the Hellinger distance while 72.6% and 72.9% respectively for $L_1$ and $L_2$. Based on the general results, at least one condition can be found where AICr performs better compared to any other methods.

If the tables are examined in terms of distributions, AICr showed its superiority in almost every condition on $t$ based on $L_1$ and $L_2$. The only exception for $t$ distribution is TS method

**Table 1. The AICr column reports the average Hellinger distance while other reports the relative Hellinger distances compared to AICr.**

| distr | n | AICr | B | R | CV | Bc | Rc | CVc | AIC | BIC | TS |
|---|---|---|---|---|---|---|---|---|---|---|---|
| 0.5N(0,1)+0.5N(4,1) | 50 | 0.052 | 1.032 | 1.015 | 1.283 | 1.015 | 1.010 | 0.942 | 1.767 | 1.294 | 1.031 |
| 0.5N(0,1)+0.5N(4,1) | 100 | 0.035 | 1.242 | 1.187 | 1.569 | 1.231 | 1.187 | 0.933 | 2.303 | 1.300 | 1.100 |
| 0.5N(0,1)+0.5N(4,1) | 200 | 0.023 | 1.198 | 1.133 | 2.045 | 1.186 | 1.130 | 0.911 | 3.049 | 1.283 | 0.807 |
| 0.5N(0,1)+0.5N(4,1) | 400 | 0.017 | 1.060 | 1.015 | 2.503 | 1.052 | 1.013 | 0.868 | 3.449 | 1.195 | 0.692 |
| 0.5N(0,1)+0.5N(4,1) | 800 | 0.012 | 0.972 | 0.940 | 2.587 | 0.968 | 0.939 | 0.834 | 3.250 | 1.051 | 0.582 |
| B(1.5,1.5) | 50 | 0.038 | 0.713 | 0.705 | 1.555 | 0.695 | 0.702 | 0.957 | 2.251 | 1.313 | 0.718 |
| B(1.5,1.5) | 100 | 0.024 | 0.758 | 0.759 | 2.094 | 0.746 | 0.756 | 1.056 | 3.174 | 1.350 | 0.759 |
| B(1.5,1.5) | 200 | 0.016 | 0.876 | 0.862 | 2.827 | 0.870 | 0.861 | 1.142 | 4.311 | 1.308 | 0.879 |
| B(1.5,1.5) | 400 | 0.011 | 0.853 | 0.801 | 3.529 | 0.850 | 0.800 | 1.203 | 4.931 | 1.146 | 0.901 |
| B(1.5,1.5) | 800 | 0.008 | 0.661 | 0.648 | 3.516 | 0.658 | 0.648 | 1.187 | 4.492 | 0.933 | 0.607 |
| exp(1) | 50 | 0.048 | 1.179 | 1.099 | 1.353 | 1.130 | 1.101 | 0.857 | 1.865 | 1.319 | 0.837 |
| exp(1) | 100 | 0.032 | 1.137 | 1.079 | 1.664 | 1.081 | 1.066 | 0.786 | 2.492 | 1.309 | 0.739 |
| exp(1) | 200 | 0.021 | 1.098 | 1.058 | 2.191 | 1.063 | 1.048 | 0.739 | 3.281 | 1.245 | 0.659 |
| exp(1) | 400 | 0.015 | 1.005 | 0.984 | 2.740 | 0.987 | 0.980 | 0.694 | 3.732 | 1.156 | 0.581 |
| exp(1) | 800 | 0.011 | 0.895 | 0.882 | 2.822 | 0.884 | 0.878 | 0.655 | 3.615 | 1.006 | 0.493 |
| N(0,1) | 50 | 0.047 | 1.215 | 1.138 | 1.386 | 1.199 | 1.139 | 0.943 | 1.930 | 1.327 | 1.233 |
| N(0,1) | 100 | 0.032 | 1.115 | 1.064 | 1.707 | 1.102 | 1.063 | 0.896 | 2.479 | 1.305 | 0.943 |
| N(0,1) | 200 | 0.022 | 1.136 | 1.091 | 2.162 | 1.132 | 1.089 | 0.861 | 3.225 | 1.251 | 0.802 |
| N(0,1) | 400 | 0.016 | 1.036 | 0.992 | 2.616 | 1.030 | 0.991 | 0.825 | 3.584 | 1.137 | 0.737 |
| N(0,1) | 800 | 0.012 | 0.924 | 0.886 | 2.606 | 0.920 | 0.885 | 0.783 | 3.326 | 1.004 | 0.636 |
| t(3) | 50 | 0.056 | 1.202 | 1.139 | 1.281 | 1.213 | 1.183 | 1.059 | 1.689 | 1.262 | 0.937 |
| t(3) | 100 | 0.038 | 1.232 | 1.189 | 1.524 | 1.275 | 1.245 | 0.995 | 2.164 | 1.273 | 0.819 |
| t(3) | 200 | 0.026 | 1.206 | 1.177 | 1.927 | 1.262 | 1.235 | 0.965 | 2.834 | 1.272 | 0.752 |
| t(3) | 400 | 0.018 | 1.166 | 1.149 | 2.373 | 1.216 | 1.200 | 0.887 | 3.180 | 1.239 | 0.699 |
| t(3) | 800 | 0.012 | 1.108 | 1.114 | 2.553 | 1.149 | 1.149 | 0.809 | 3.163 | 1.111 | 0.629 |
| U(0,1) | 50 | 0.037 | 0.587 | 0.551 | 1.616 | 0.542 | 0.547 | 0.930 | 2.327 | 1.309 | 0.576 |
| U(0,1) | 100 | 0.022 | 0.479 | 0.460 | 2.243 | 0.453 | 0.457 | 1.081 | 3.402 | 1.248 | 0.480 |
| U(0,1) | 200 | 0.014 | 0.363 | 0.355 | 3.277 | 0.350 | 0.353 | 1.273 | 5.013 | 1.166 | 0.380 |
| U(0,1) | 400 | 0.009 | 0.246 | 0.243 | 4.469 | 0.241 | 0.242 | 1.483 | 6.195 | 1.000 | 0.256 |
| U(0,1) | 800 | 0.006 | 0.119 | 0.119 | 4.917 | 0.118 | 0.119 | 1.610 | 6.316 | 0.761 | 0.128 |

**Table 2. The AICr column reports the average $L_1$ distance while other reports the relative $L_1$ distances compared to AICr.**

| distr | n | AICr | B | R | CV | Bc | Rc | CVc | AIC | BIC | TS |
|---|---|---|---|---|---|---|---|---|---|---|---|
| 0.5N(0,1)+0.5N(4,1) | 50 | 0.407 | 1.075 | 1.067 | 1.224 | 1.065 | 1.062 | 0.972 | 1.396 | 1.176 | 1.072 |
| 0.5N(0,1)+0.5N(4,1) | 100 | 0.332 | 1.265 | 1.230 | 1.390 | 1.261 | 1.229 | 1.017 | 1.616 | 1.168 | 1.135 |
| 0.5N(0,1)+0.5N(4,1) | 200 | 0.270 | 1.219 | 1.172 | 1.622 | 1.215 | 1.170 | 1.048 | 1.895 | 1.157 | 0.927 |
| 0.5N(0,1)+0.5N(4,1) | 400 | 0.226 | 1.112 | 1.089 | 1.799 | 1.110 | 1.087 | 1.057 | 2.018 | 1.123 | 0.852 |
| 0.5N(0,1)+0.5N(4,1) | 800 | 0.191 | 1.097 | 1.082 | 1.777 | 1.096 | 1.082 | 1.066 | 1.905 | 1.075 | 0.777 |
| B(1.5,1.5) | 50 | 0.290 | 0.696 | 0.700 | 1.499 | 0.688 | 0.697 | 1.092 | 1.795 | 1.156 | 0.705 |
| B(1.5,1.5) | 100 | 0.241 | 0.838 | 0.842 | 1.771 | 0.834 | 0.841 | 1.219 | 2.118 | 1.173 | 0.850 |
| B(1.5,1.5) | 200 | 0.202 | 0.997 | 0.982 | 2.060 | 0.997 | 0.983 | 1.307 | 2.455 | 1.116 | 1.020 |
| B(1.5,1.5) | 400 | 0.173 | 0.980 | 0.937 | 2.260 | 0.980 | 0.937 | 1.369 | 2.563 | 1.017 | 1.028 |
| B(1.5,1.5) | 800 | 0.151 | 0.852 | 0.846 | 2.103 | 0.852 | 0.845 | 1.345 | 2.276 | 0.922 | 0.752 |
| exp(1) | 50 | 0.359 | 1.261 | 1.222 | 1.351 | 1.230 | 1.215 | 0.993 | 1.544 | 1.243 | 0.945 |
| exp(1) | 100 | 0.290 | 1.266 | 1.244 | 1.547 | 1.230 | 1.229 | 0.970 | 1.806 | 1.234 | 0.883 |
| exp(1) | 200 | 0.236 | 1.275 | 1.267 | 1.817 | 1.252 | 1.255 | 0.963 | 2.144 | 1.219 | 0.837 |
| exp(1) | 400 | 0.194 | 1.258 | 1.263 | 2.066 | 1.247 | 1.258 | 0.960 | 2.328 | 1.185 | 0.793 |
| exp(1) | 800 | 0.162 | 1.231 | 1.240 | 2.048 | 1.223 | 1.236 | 0.969 | 2.201 | 1.139 | 0.742 |
| N(0,1) | 50 | 0.353 | 1.235 | 1.175 | 1.360 | 1.227 | 1.176 | 1.042 | 1.571 | 1.222 | 1.259 |
| N(0,1) | 100 | 0.294 | 1.137 | 1.107 | 1.551 | 1.131 | 1.105 | 1.046 | 1.810 | 1.202 | 0.995 |
| N(0,1) | 200 | 0.242 | 1.193 | 1.167 | 1.786 | 1.192 | 1.166 | 1.071 | 2.099 | 1.181 | 0.886 |
| N(0,1) | 400 | 0.203 | 1.151 | 1.129 | 1.982 | 1.150 | 1.129 | 1.088 | 2.219 | 1.130 | 0.834 |
| N(0,1) | 800 | 0.172 | 1.121 | 1.102 | 1.923 | 1.121 | 1.101 | 1.098 | 2.065 | 1.069 | 0.764 |
| t(3) | 50 | 0.401 | 1.189 | 1.156 | 1.283 | 1.199 | 1.181 | 1.071 | 1.441 | 1.194 | 1.023 |
| t(3) | 100 | 0.323 | 1.247 | 1.227 | 1.463 | 1.272 | 1.260 | 1.081 | 1.680 | 1.209 | 0.919 |
| t(3) | 200 | 0.262 | 1.257 | 1.254 | 1.697 | 1.294 | 1.289 | 1.103 | 1.971 | 1.212 | 0.885 |
| t(3) | 400 | 0.213 | 1.282 | 1.291 | 1.932 | 1.316 | 1.323 | 1.109 | 2.149 | 1.217 | 0.862 |
| t(3) | 800 | 0.172 | 1.293 | 1.324 | 1.972 | 1.327 | 1.350 | 1.103 | 2.109 | 1.209 | 0.827 |
| U(0,1) | 50 | 0.277 | 0.307 | 0.307 | 1.552 | 0.293 | 0.302 | 1.048 | 1.880 | 1.068 | 0.321 |
| U(0,1) | 100 | 0.214 | 0.202 | 0.205 | 1.939 | 0.196 | 0.202 | 1.262 | 2.358 | 0.971 | 0.217 |
| U(0,1) | 200 | 0.169 | 0.126 | 0.130 | 2.418 | 0.124 | 0.128 | 1.495 | 2.922 | 0.890 | 0.151 |
| U(0,1) | 400 | 0.136 | 0.078 | 0.080 | 2.840 | 0.078 | 0.079 | 1.693 | 3.232 | 0.767 | 0.091 |
| U(0,1) | 800 | 0.110 | 0.048 | 0.050 | 2.816 | 0.048 | 0.050 | 1.770 | 3.062 | 0.628 | 0.059 |

even though AICr has better performance for a small sample size. Similarly, the AICr method is best for 0.5N(0,1)+0.5N(4,1), N(0,1) and exponential distributions for the small and moderate sample sizes. For these distributions, AICr performed weakly only against CVc and TS based on the Hellinger distance. This situation mostly changed for the $L_1$ and $L_2$ distances. AICr is more successful for small samples when compared to TS while CVc only stayed strong for the exponential distribution. If the U(0,1) and the B(1.5,1.5) distributions are considered in general, the results of these distributions are close to each other and AICr performed well compared with CV, CVc, AIC and BIC. However, the noteworthy point here is that AICr's estimated the density at least 2 times better than these methods. The distances of AIC and BIC methods reach maximum values for all three tables and it is followed by CV. This is expected for only AIC as [9, 26, 39] indicated that AIC underpenalize even for regular histograms. To sum up, these methods are weakest against AICr based on the Monte Carlo results. On the contrary, TS is one of the most successful methods and it is followed by CVc if we consider the Hellinger distance. As can be observed in general, AICr typically performs better on small and moderate sample sizes, whereas the success of the other approaches varies depending on the distribution and sample size. Conversely, AICr occasionally performed poorly at large sample sizes.

**Table 3. The AICr column reports the average $L_2$ distance while other reports the relative $L_2$ distances compared to AICr.**

| distr | n | AICr | B | R | CV | Bc | Rc | CVc | AIC | BIC | TS |
|---|---|---|---|---|---|---|---|---|---|---|---|
| 0.5N(0,1)+0.5N(4,1) | 50 | 0.028 | 1.107 | 1.002 | 2.790 | 0.974 | 0.974 | 0.957 | 15.273 | 6.732 | 1.050 |
| 0.5N(0,1)+0.5N(4,1) | 100 | 0.019 | 1.429 | 1.309 | 3.737 | 1.341 | 1.292 | 1.013 | 44.671 | 4.159 | 1.232 |
| 0.5N(0,1)+0.5N(4,1) | 200 | 0.013 | 1.397 | 1.255 | 5.302 | 1.324 | 1.246 | 1.057 | 153.131 | 3.314 | 0.850 |
| 0.5N(0,1)+0.5N(4,1) | 400 | 0.010 | 1.141 | 1.076 | 6.789 | 1.109 | 1.070 | 1.061 | 67.384 | 7.027 | 0.666 |
| 0.5N(0,1)+0.5N(4,1) | 800 | 0.007 | 1.043 | 1.011 | 6.658 | 1.030 | 1.008 | 1.065 | 37.281 | 2.834 | 0.522 |
| B(1.5,1.5) | 50 | 0.151 | 0.637 | 0.474 | 4.090 | 0.450 | 0.465 | 1.053 | 34.513 | 16.774 | 0.672 |
| B(1.5,1.5) | 100 | 0.106 | 0.700 | 0.620 | 6.615 | 0.597 | 0.611 | 1.269 | 113.643 | 19.338 | 0.652 |
| B(1.5,1.5) | 200 | 0.079 | 0.847 | 0.790 | 7.116 | 0.804 | 0.790 | 1.414 | 61.436 | 12.289 | 0.835 |
| B(1.5,1.5) | 400 | 0.060 | 0.817 | 0.728 | 8.186 | 0.787 | 0.727 | 1.496 | 60.910 | 10.639 | 0.851 |
| B(1.5,1.5) | 800 | 0.049 | 0.573 | 0.550 | 7.362 | 0.560 | 0.549 | 1.425 | 35.979 | 5.934 | 0.468 |
| exp(1) | 50 | 0.072 | 2.477 | 1.423 | 4.866 | 1.355 | 1.348 | 1.020 | 45.877 | 27.386 | 1.205 |
| exp(1) | 100 | 0.046 | 1.910 | 1.472 | 6.742 | 1.385 | 1.409 | 0.987 | 64.935 | 15.813 | 1.060 |
| exp(1) | 200 | 0.030 | 1.723 | 1.518 | 9.833 | 1.434 | 1.472 | 0.959 | 199.342 | 8.611 | 0.844 |
| exp(1) | 400 | 0.020 | 1.583 | 1.513 | 12.317 | 1.432 | 1.493 | 0.943 | 145.737 | 6.164 | 0.723 |
| exp(1) | 800 | 0.014 | 1.468 | 1.465 | 12.359 | 1.380 | 1.446 | 0.938 | 72.781 | 3.894 | 0.622 |
| N(0,1) | 50 | 0.039 | 1.606 | 1.285 | 3.946 | 1.363 | 1.271 | 1.058 | 26.257 | 10.123 | 1.573 |
| N(0,1) | 100 | 0.027 | 1.311 | 1.139 | 5.292 | 1.185 | 1.131 | 1.084 | 67.288 | 9.027 | 0.941 |
| N(0,1) | 200 | 0.019 | 1.303 | 1.211 | 7.462 | 1.250 | 1.206 | 1.139 | 159.647 | 6.187 | 0.690 |
| N(0,1) | 400 | 0.014 | 1.166 | 1.100 | 9.220 | 1.132 | 1.097 | 1.163 | 49.997 | 2.906 | 0.573 |
| N(0,1) | 800 | 0.010 | 1.055 | 1.004 | 8.760 | 1.031 | 1.003 | 1.170 | 37.379 | 1.889 | 0.453 |
| t(3) | 50 | 0.031 | 1.811 | 1.343 | 3.940 | 1.416 | 1.379 | 1.154 | 29.200 | 8.119 | 1.388 |
| t(3) | 100 | 0.021 | 1.677 | 1.496 | 5.461 | 1.579 | 1.563 | 1.209 | 66.353 | 8.259 | 0.869 |
| t(3) | 200 | 0.013 | 1.669 | 1.585 | 8.037 | 1.670 | 1.678 | 1.290 | 122.217 | 8.370 | 0.789 |
| t(3) | 400 | 0.009 | 1.725 | 1.741 | 11.655 | 1.796 | 1.844 | 1.353 | 59.854 | 4.067 | 0.741 |
| t(3) | 800 | 0.005 | 1.804 | 1.927 | 13.112 | 1.897 | 2.017 | 1.382 | 54.284 | 2.688 | 0.655 |
| U(0,1) | 50 | 0.164 | 0.475 | 0.288 | 3.872 | 0.266 | 0.274 | 0.940 | 36.447 | 15.831 | 0.451 |
| U(0,1) | 100 | 0.108 | 0.283 | 0.219 | 5.770 | 0.202 | 0.207 | 1.181 | 54.243 | 18.418 | 0.278 |
| U(0,1) | 200 | 0.075 | 0.174 | 0.148 | 8.338 | 0.140 | 0.143 | 1.414 | 97.139 | 38.224 | 0.193 |
| U(0,1) | 400 | 0.053 | 0.105 | 0.101 | 9.207 | 0.098 | 0.099 | 1.643 | 95.629 | 16.172 | 0.120 |
| U(0,1) | 800 | 0.038 | 0.069 | 0.070 | 8.877 | 0.068 | 0.069 | 1.750 | 78.499 | 1.912 | 0.081 |

## Empirical examples

This section presents the comparison of the AICr method with the other methods (B, R, CV, Bc, Rc, CVc, AIC, BIC, TS) through some real empirical data sets. There are six datasets chosen to illustrate different features often encountered in real applications from the histogram literature. The datasets represented a wide range of sample sizes, and their references are given in Table 4. Among these data sets, especially Weisberg's (1980) study has been widely used to analyze histogram bin widths [40], e.g. Azzalini and Bowman (1990), Davies et al. (2009), Scott (2015), Silverman (2018) and Li et al. (2020) [2, 10, 41–43]. Similarly, the Buffalo snowfall data was also analyzed in e.g. Atilgan (1990), Scott (2015) and Silverman (2018) [2, 20, 42] and it was updated by using data from US National Weather Service in 2021 (https://www.weather.gov/buf/BuffaloSnow). The suicidal risk data set was also used in Davies et al. (2009) and Silverman (2018) [2, 10]. The last data set is from Scott et al. (1978) and has two variables for 371 male patients; concentration of plasma cholesterol and plasma triglycerides (mg/dl) [44]. There are two groups of patients that depend on whether those with narrowing of the arteries or not and in this present study we use only the 320 patients without narrowing (for detail see Table B3 in [42].).

**Table 4. The determined amount of bins for different data-based methods through real data set.**

| Variable | n | Reference |
|---|---:|---:|
| Geyser dur. | 299 | [40] |
| Geyser wait | 299 | [40] |
| B. snowfall | 112 | [45] |
| Suicidal | 86 | [46] |
| Cholesterol | 320 | [44] |
| Triglycerides | 320 | [44] |

The main problem with the empirical analysis is that we do not know the true underlying distribution. However, we can interpret the trade-off between being able to detect valuable patterns and patterns arising from noise. In Fig 1, the number of bins change between 6 to 22 while the proposed AICr method has 15. It can be seen from the figure that, there are two peaks and AICr has captured one of them, the one on the right. This peak was captured by almost every method, but the same cannot be said for the decay to the right side of the right peak. B, R, Bc, Rc can not detect this decay and all of them have bins lower than 10, which is defined as oversmoothing problem. Also, TS has the maximum bin number in Fig 1 which is the same as for Fig 2 (22 and 34 bins respectively). This gives rise to the suspicion of undersmoothing as it seems too random. This is followed by AIC with 18 bins. On the other hand, B, R, Bc, Rc and BIC, which have 5 and 6 bins, seem oversmoothing. Besides, even though both AICr and CV have the almost same amount of bins, CV shows the fluctuations better and the situation of CVc is similar to CV with just 11 bins.

The snowfall data is also an interesting example for evaluating methods since there is an outlier on the right-hand side at about the 199th order of data. It means that if there are too few bins the outlier will be connected with other large observations. This happened for every method in Fig 3 except the proposed AICr. It is an important finding that, for all methods except AICr, it seems like there are some patterns displaying that are not really there between the range of (150–200). Also, B, R, Bc, Rc, BIC and TS hide the shape of distribution since they have fewer bins. The suicidal data which can be seen in Fig 4 is an example of a heavily skewed distribution. It means that the fewer bins imply doesn't enough reveal how the data looks like for the left-hand side and oversmooth on the right-hand side, like B, R, Bc and Rc. On the contrary, CV and AIC display too many going up and down patterns with 12 and 13 bins respectively. According to this, TS with 10 bins, brings the question that whether the distribution first goes up and then down as for AICr or down all the time as for CVc and BIC.

The cholesterol data can be taken as a representative of symmetrical data, with a few outliers to the right side, see Fig 5. As happened above, too few bins (such as those given by B, R, Bc, Rc and BIC) display a nonrandom shape while too many bins (such as CV and AIC) display the distribution too random. Further, in AIC and CV, they appear as if there are some patterns that aren't actually there because of outliers in the 300 to 400 range. Also, they have very small fluctuating bins at around 220 and this leads to an unsymmetrical pattern. However, it is hard to know that these movements are due to a lower density in this region of the data. TS also doesn't seem to reveal enough information although it has 15 bins and is similar to AICr with 8 bins. For this data set, CVc looks like fairly good to catch the symmetrical patterns with 9 bins and it is closer to the AICr. Finally, the triglycerides concentration data, in Fig 6, is slightly positively skewed. The methods B, R, Bc, Rc and BIC (5 and 6 bins) have too few bins to reveal the underlying pattern of the data. TS has the same patterns even though it has 19 bins. Surprisingly, as above, TS has almost the same patterns with histograms that have too few bins.

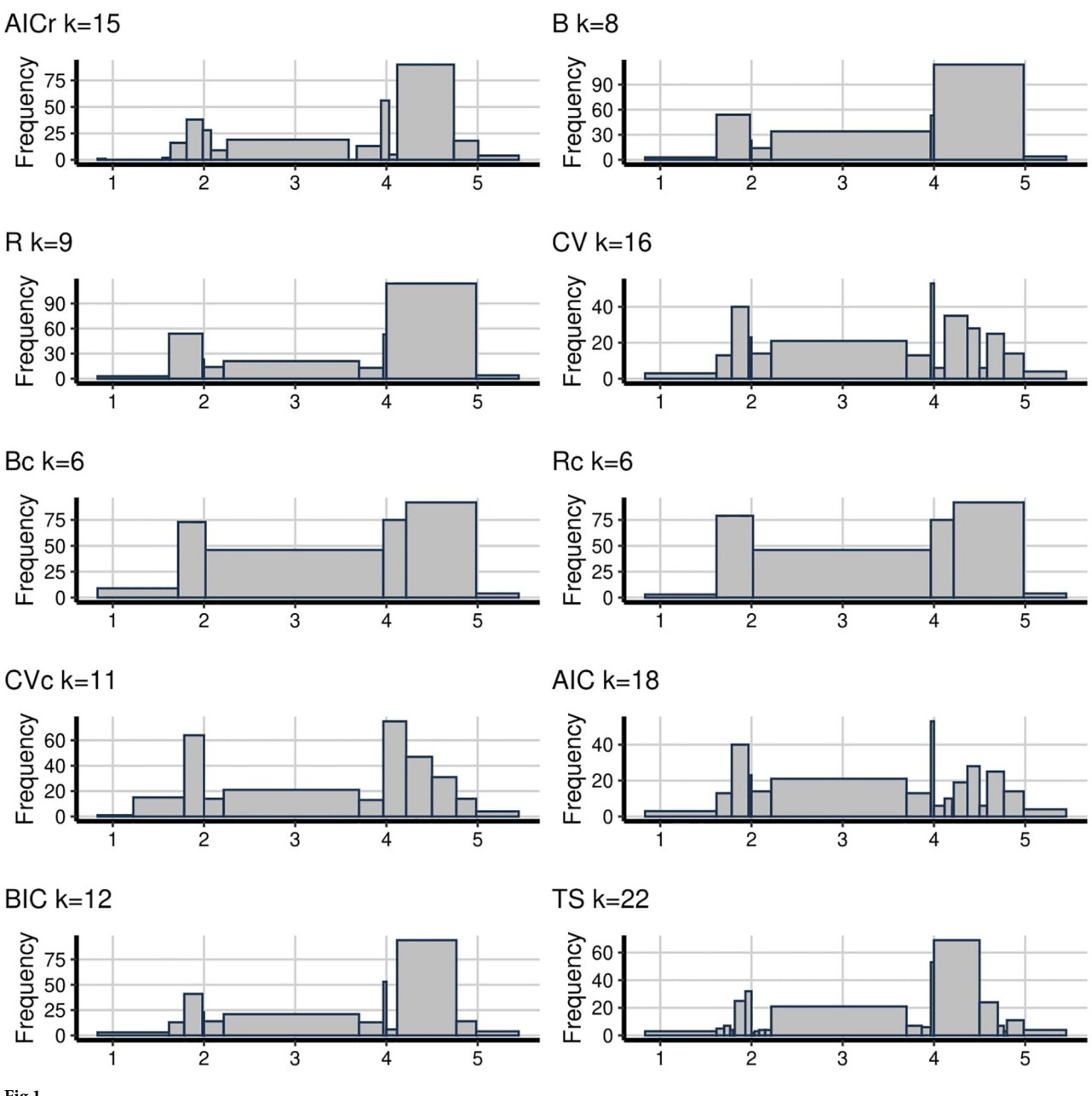

**Fig 1.**

Also, CV and AIC look too random since they have too many bins, 22 and 25 respectively. For these data examples, AICr and CVc seem like doing a fairly good job (AICr has 14 while CVc has 8 bins). The only difference between them, AICr has a decreasing move in the range 150 to 250. But again, it is hard to know whether this movement is due to whether there is a lower density in this region of the data.

To sum up, it can be shown from the examples too, that deciding the number of bins is an important issue. Undersmoothing can show noisy results while oversmoothing can hide some interesting clues about data. In general, B, R, Bc, Rc and BIC tend to produce histograms with very few bins while CV and AIC produce them with too many bins. Also, TS is competing

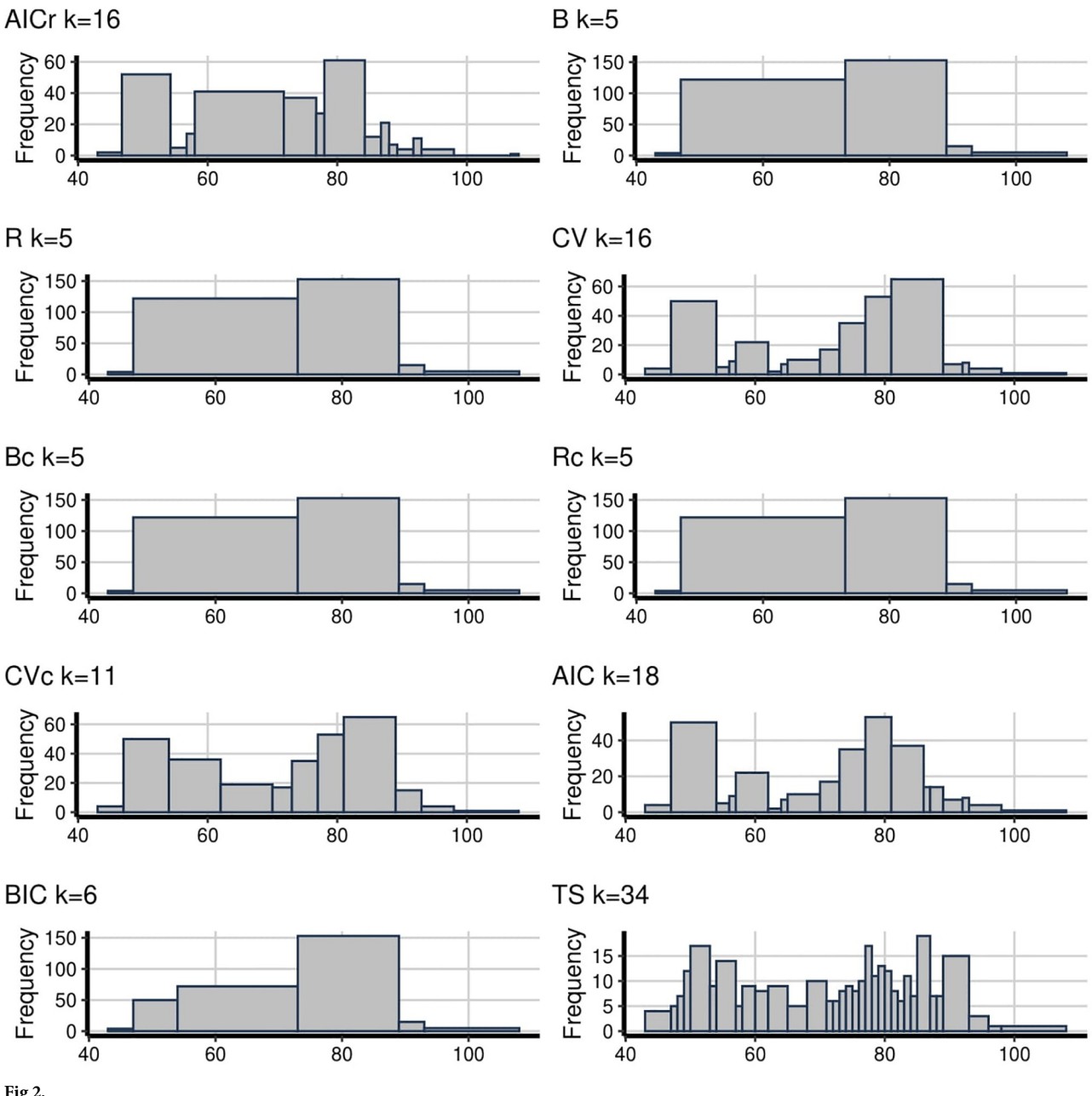

**Fig 2.**

with the methods that have too many bins and the histograms of TS looks displays similar patterns to the rough histograms that have few bins. In these methods, only AICr and CVc generate bin numbers between these two extreme points and AICr often tends to have higher bins than CVc.

## Conclusions

In this study, we introduced a procedure, based on the very well-known AIC, for histograms that have unequal bin width. Therefore, after the introduction of the restricted likelihood

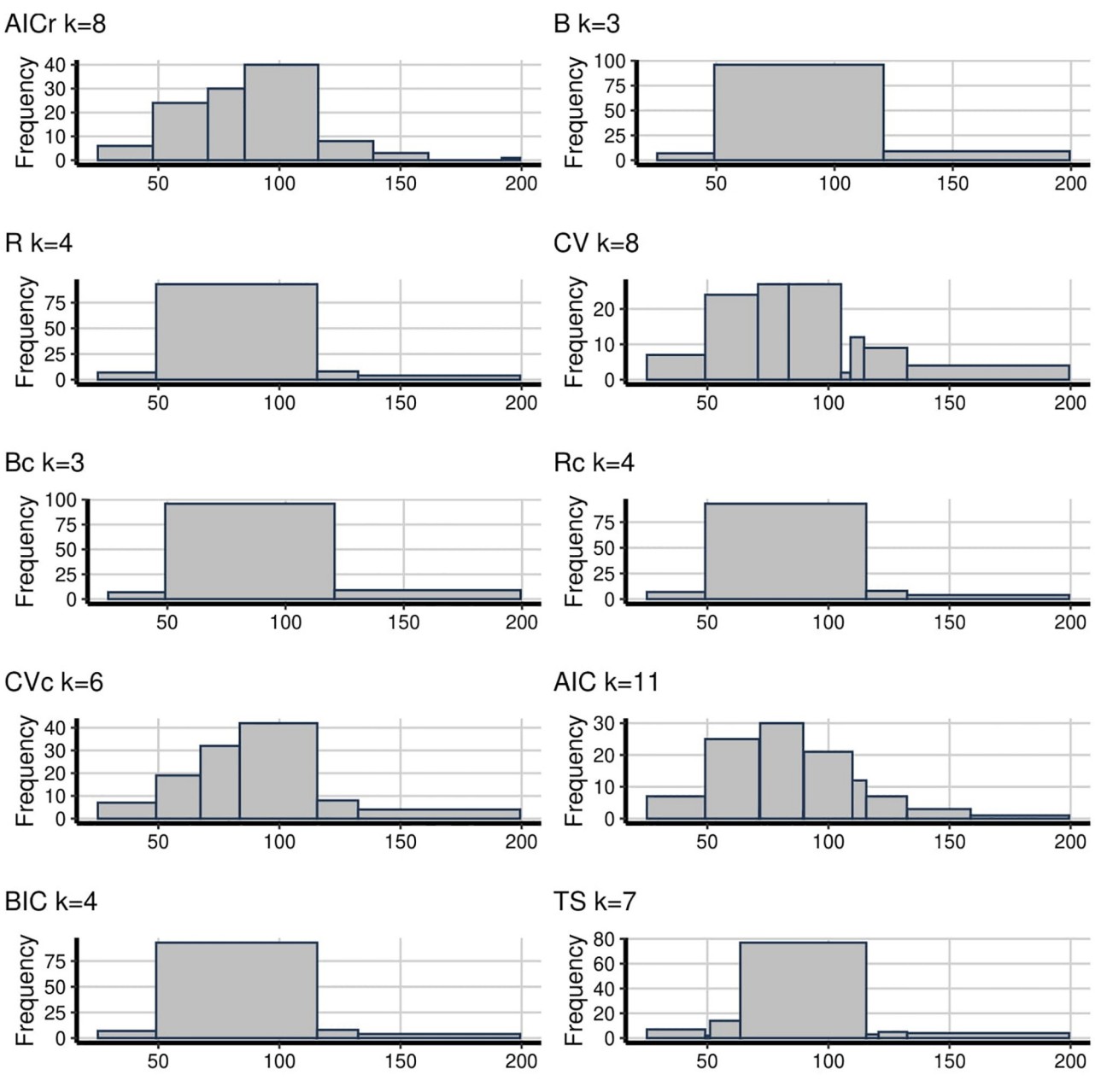

**Fig 3.**

based AIC (AICr) in Section for AICr, the performance of AICr is compared with 9 other methods that are used by [26] for unequal histograms through both Monte Carlo simulations and using empirical data. The evaluation of methods is made by calculating distances between estimated and its true ones e.g. Hellinger distance.

Data is generated from a wide range of distributions for the simulation study: Normal, half-normal, exponential, uniform, *t* and Beta. Also, sample sizes are arranged to include a wide range: 50, 100, 200, 400 and 800. According to the simulation results, the success rate of AICr method is over 60% for each distance. It has been observed that AICr performs best, especially

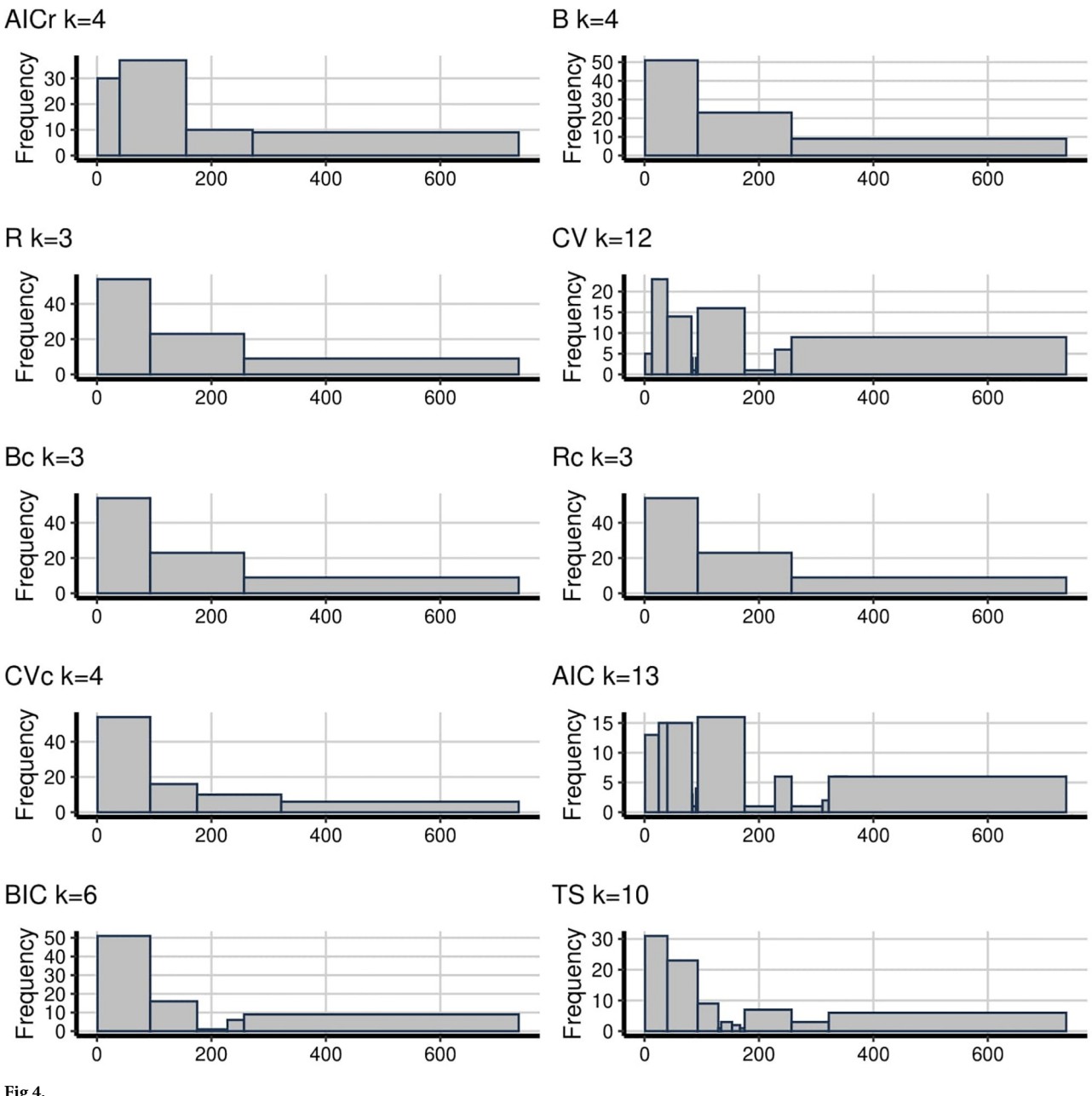

**Fig 4.**

for the *t*, exponential, normal and half-normal distributions and it is superior for small and moderate sample sizes. On the contrary, the B(1.5,1.5) and U(0,1) distributions are the only ones where AICr is often weak. For these distributions, AIC and CV's have the worst performance followed by BIC among all the methods used in this study. The method denoted as TS performs well for moderate and large sample sizes while it performs worse for small sample sizes.

In the empirical part of the paper, six datasets are used to exemplify the different features of methods. The AICr method performs best among all methods as it is neither oversmooth nor undersmooth. The B, R, Bc, Rc and BIC methods tend to be oversmoothing since they have

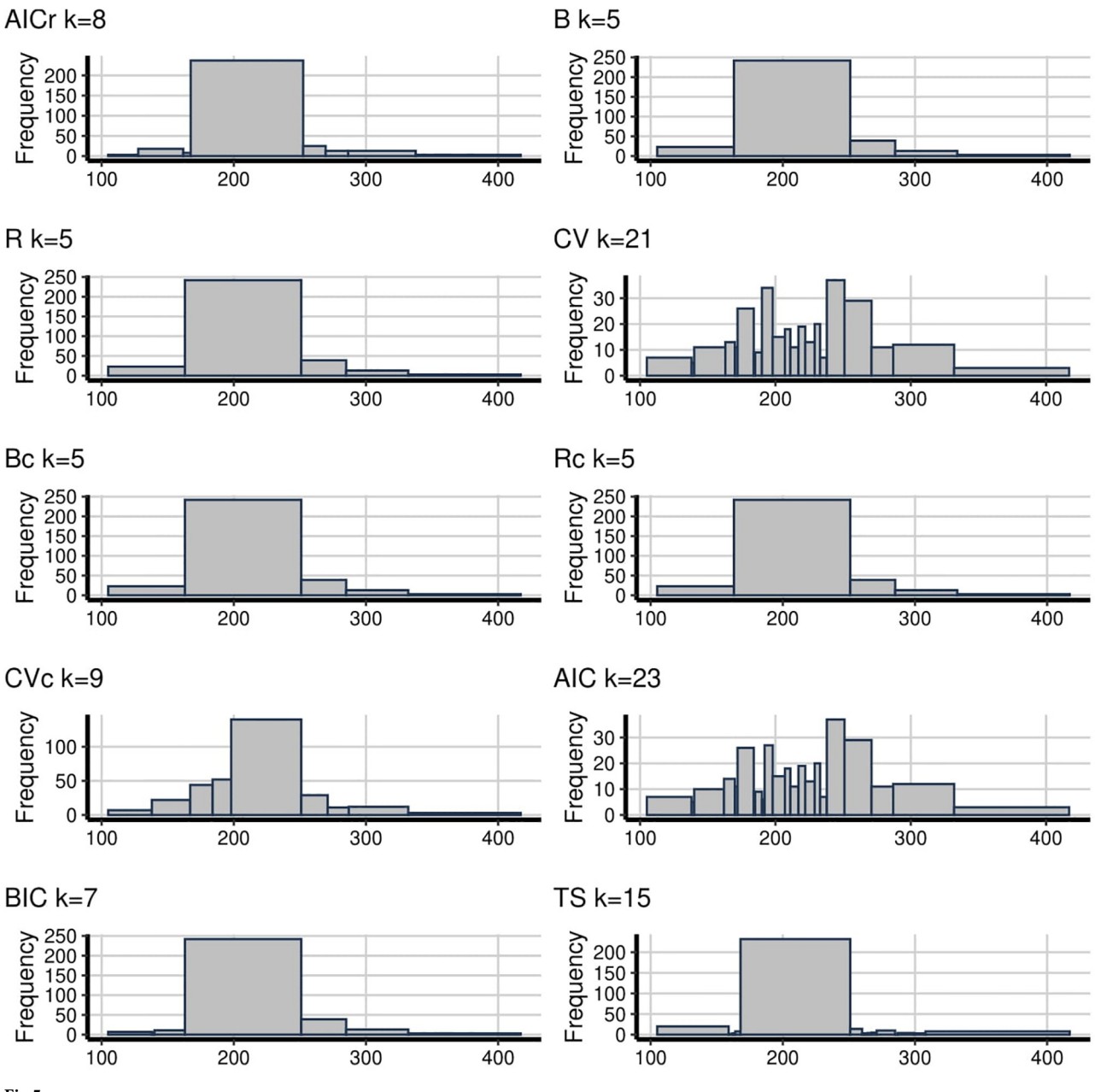

**Fig 5.**

few bins while CV and AIC tend to be undersmoothing with too many bins. One of the important findings is that TS tends to produce histograms with a high amount of bins however they display almost the same patterns with fewer bin histograms, which means it can be hard to detect important information of data. This result produces a devastating effect against the positive results of TS in the simulations. Only two methods, AICr and CVc, choose a moderate amount of bins for the histograms and avoid histograms that are too noisy or too rough. Further, only AICr could catch the outlier in the snowfall data.

In brief, the proposed AICr method performs very well according to both simulations and empirical examples. The performance of the method shows that AICr can be used to

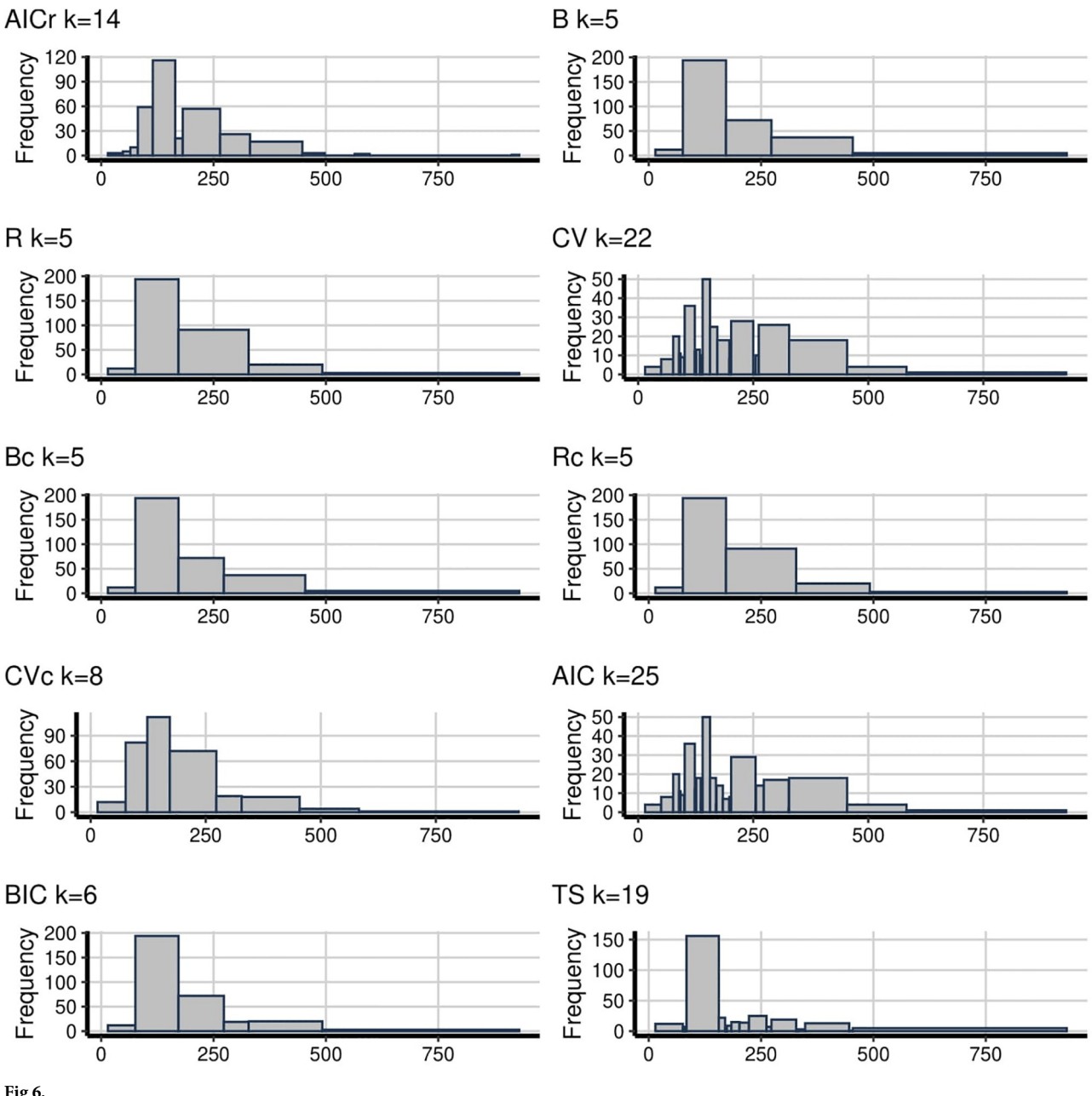

**Fig 6.**

determine the bin size of unequal histograms. Future studies are needed to get a more complete understanding of the situations when the different methods perform best.

## Supporting information

**S1 File.**
(PDF)

## Author Contributions

**Data curation:** Johan Lyhagen.

**Methodology:** Johan Lyhagen.

**Resources:** Sahika Gokmen.

**Visualization:** Sahika Gokmen.

**Writing – original draft:** Sahika Gokmen.

**Writing – review & editing:** Sahika Gokmen.

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
