## [Decision Letter · Decision Letter 0]

16 May 2023

PONE-D-23-09990The performance of restricted AIC for irregular histogram modelsPLOS ONE

Dear Dr. GÖKMEN,

Thank you for submitting your manuscript to PLOS ONE. After careful consideration, we feel that it has merit but does not fully meet PLOS ONE’s publication criteria as it currently stands. Therefore, we invite you to submit a revised version of the manuscript that addresses the points raised during the review process.

This paper proposes a novel approach to AIC for histograms with unequal bin widths. Extensive Monte Carlo simulations and empirical examples are provided to demonstrate the applicability of the suggested approach. Overall, the paper's main idea is interesting. However, some comments and suggestions pointed out by reviewers should be addressed accordingly and need further clarification to enhance the readability purposes. 

We look forward to receiving your revised manuscript.

Kind regards,

Kok Haur Ng, Ph.D.

Academic Editor

PLOS ONE

Journal Requirements:

 "Şahika Gökmen is supported by the International Postdoctoral Research Scholarship Program of The Scientific and Technological Research Council of Turkey (TUBITAK BIDEB 2219)."

"Sahika Gokmen is supported by the International Postdoctoral Research Scholarship 280

Program of The Scientific and Technological Research Council of Turkey (TUBITAK 281

BIDEB 2219)."

"Şahika Gökmen is supported by the International Postdoctoral Research Scholarship Program of The Scientific and Technological Research Council of Turkey (TUBITAK BIDEB 2219)."

7. In your Data Availability statement, you have not specified where the minimal data set underlying the results described in your manuscript can be found. PLOS defines a study's minimal data set as the underlying data used to reach the conclusions drawn in the manuscript and any additional data required to replicate the reported study findings in their entirety. All PLOS journals require that the minimal data set be made fully available. For more information about our data policy, please see http://journals.plos.org/plosone/s/data-availability.

Additional Editor Comments:

We have completed the review of your manuscript. Both of the reviewers have recommended a minor revision. When revising your manuscript, please consider all issues mentioned in the reviewers' comments, outline every change made in response to their comments and provide suitable rebuttals for any comments not addressed.

Reviewers' comments:

Reviewer's Responses to Questions

**Comments to the Author**

1. Is the manuscript technically sound, and do the data support the conclusions?

Reviewer #1: Partly

Reviewer #2: Yes

2. Has the statistical analysis been performed appropriately and rigorously? 

Reviewer #1: Yes

Reviewer #2: Yes

3. Have the authors made all data underlying the findings in their manuscript fully available?

Reviewer #1: Yes

Reviewer #2: Yes

4. Is the manuscript presented in an intelligible fashion and written in standard English?

Reviewer #1: Yes

Reviewer #2: Yes

5. Review Comments to the Author

Reviewer #1: In this study, a novel approach to AIC for histograms with unequal bin widths has been proposed by authors and demonstrated the advantage of the proposed approach in comparison to others using both extensive Monte Carlo simulations and empirical examples. It worthy contribution. However I have the following comments.

1)In the paper, authors may clearly described the novelty and main contributions with justification both technically and scientifically.

2) The authors need to provide solid background for research gap keeping in view most recently published articles.

3) The authors also need discuss the shortcomings of the study performed in this paper.

4) The literature review is short and should add some recently and more relavent published with main focus on the title of the paper.

Reviewer #2: The manuscript entitled " The Performance of restricted AIC for irregular histogram models ", the author has proposed a modified approach to using AIC for histograms with different bin widths. The manuscript gives a comprehensive analysis and discussion on the proposed AIC method. There are lots of grammatical errors all over the manuscript. I suggest the author send the manuscript for proofreading.

Some minor corrections:

a. Line 45: Barron et al. (1999) develops ---- Barron et al. (1999) developed

b. All mathematics equations should be written in complete sentences, so equations must have punctuation.

C. Line 108: tables 1, 2, and 3 ---- Table 1, 2 and 3.

d. Line 109: Table 2 and 3 --- Tables 2 and 3

e. Line 195: Further , in AIC and CV, it appears----- Further, in AIC and Cv, they appear

f. Line 201: ... 9 bins and is is closer --- 9bins and it is closer

g. Line 207: AICr and CVc seems like --- Line 207: AICr and CVr seem like

6. PLOS authors have the option to publish the peer review history of their article (what does this mean?). If published, this will include your full peer review and any attached files.

Reviewer #1: **Yes: **Gajendra K. Vishwakarma

Reviewer #2: No

---

## [Author Response · Author response to Decision Letter 0]

30 Jun 2023

To Reviewers:

First of all, we want to thank the reviewers for substantially improving the manuscript! Please find each of the comments below with our response.

Reviewer 1:

1- Based on the suggestion of the reviewer, the novelty, research gap and the main contributions of the paper are rewritten in the Introduction section.

2- We searched the literature again and haven’t found anything more recent about the related field.

3- The weak performances of proposed AICr are implied both in Monte Carlo and Conclusion sections.

4- Again, unfortunately we searched the literature again and haven’t found anything more recent about the related field.

Reviewer 2:

Grammatical errors are corrected in the text.

---

## [Decision Letter · Decision Letter 1]

26 Jul 2023

The performance of restricted AIC for irregular histogram models

PONE-D-23-09990R1

Dear Dr. GÖKMEN,

We’re pleased to inform you that your manuscript has been judged scientifically suitable for publication and will be formally accepted for publication once it meets all outstanding technical requirements.

Kind regards,

Kok Haur Ng, Ph.D.

Academic Editor

PLOS ONE

Additional Editor Comments (optional):

The authors have made improvements as suggested. The revised manuscript sounds better.

Reviewers' comments:

Reviewer's Responses to Questions

**Comments to the Author**

1. If the authors have adequately addressed your comments raised in a previous round of review and you feel that this manuscript is now acceptable for publication, you may indicate that here to bypass the “Comments to the Author” section, enter your conflict of interest statement in the “Confidential to Editor” section, and submit your "Accept" recommendation.

Reviewer #1: All comments have been addressed

Reviewer #2: All comments have been addressed

2. Is the manuscript technically sound, and do the data support the conclusions?

Reviewer #1: Yes

Reviewer #2: Yes

3. Has the statistical analysis been performed appropriately and rigorously? 

Reviewer #1: Yes

Reviewer #2: Yes

4. Have the authors made all data underlying the findings in their manuscript fully available?

Reviewer #1: Yes

Reviewer #2: Yes

5. Is the manuscript presented in an intelligible fashion and written in standard English?

Reviewer #1: Yes

Reviewer #2: Yes

6. Review Comments to the Author

Reviewer #1: Authors addressed the comments in the revised version nicely, Thus it is suitable for publication without any further mofdification.

Reviewer #2: (No Response)

7. PLOS authors have the option to publish the peer review history of their article (what does this mean?). If published, this will include your full peer review and any attached files.

Reviewer #1: **Yes: **Gajendra K. Vishwakarma

Reviewer #2: No

---

## [Editor Report · Acceptance letter]

2 Aug 2023

PONE-D-23-09990R1 

The performance of restricted AIC for irregular histogram models 

Dear Dr. Gokmen:

I'm pleased to inform you that your manuscript has been deemed suitable for publication in PLOS ONE. Congratulations! Your manuscript is now with our production department. 

Kind regards, 

on behalf of

Assoc. Prof. Dr. Kok Haur Ng 

Academic Editor

PLOS ONE